# Performance evaluation of nine reference centers and comparison of DNA extraction protocols for effective surveillance of *Leishmania*-infected Phlebotomine sand flies: Basis for technical recommendations

Jorian Prudhomme[1][☉]*, Aymeric Delabarre[1][☉], Bulent Alten[2], Umut Berberoglu[3], Eduardo Berriatua[4], Gioia Bongiorno[5], José Manuel Cristovao[6], Maya Davidovich-Cohen[7], Trentina Di Muccio[5], Ozge Erisoz Kasap[2], Eleonora Fiorentino[5], Oscar D. Kirstein[7], Edwin Kniha[8], Carla Maia[6], Mesut Mungan[3], Clara Muñoz-Hernández[4,9], Muhammed Nalçaci[10], Gizem Oguz Kaskan[2], Yusuf Ozbel[10], Seray Ozensoy Toz[10], Ricardo Parreira[6], Katharina Platzgummer[8], Ceylan Polat[2], José Risueño[4], Liora Studentsky[7], Gamze Varol[3], Julia Walochnik[8], Kardelen Yetişmiş[10], Florence Robert-Gangneux[1]

**1** Univ Rennes, Inserm, EHESP, Irset (Institut de Recherche en Santé Environnement Travail), UMR S 1085, Rennes, France, **2** Hacettepe University, Ankara, Turkey, **3** Turkish Ministry of Health, Ankara, Turkey, **4** University of Murcia, Murcia, Spain, **5** Istituto Superiore di Sanità, Rome, Italy, **6** Global Health and Tropical Medicine, GHTM, LA-REAL, Instituto de Higiene e Medicina Tropical, Universidade NOVA de Lisboa, Lisbon, Portugal, **7** Jerusalem Public Health Laboratories, Ministry of Health, Jerusalem, Israel, **8** Institute of Specific Prophylaxis and Tropical Medicine, Center for Pathophysiology, Infectiology and Immunology, Medical University of Vienna, Vienna, Austria, **9** Health and Biotechnology Research Group (SaBio), Institute for Game and Wildlife Research (IREC), CSIC-UCLM-JCCM, Ciudad Real, Spain, **10** Ege University Graduate School of Natural and Applied Sciences, Department of Biology, Bornova, Izmir, Türkiye

☉ These authors contributed equally to this work.
* jorian.prudhomme@hotmail.fr

## Abstract

### Background

Leishmaniasis, caused by *Leishmania* protozoan parasites transmitted by Phlebotomine sand flies, is a significant public health concern in the Mediterranean basin. Effective monitoring of *Leishmania*-infected sand flies requires standardized tools for comparing their distribution and infection prevalence. Consistent quantitative real-time PCR (qPCR) parameters and efficient DNA extraction protocols are crucial for reliable results over time and across regions. However, the absence of standardized technical recommendations for *Leishmania* DNA detection hinders effective surveillance. This study aimed to compare different DNA extraction protocols and conduct a qPCR-based External Quality Assessment (EQA) through a multicenter study involving nine reference laboratories, with a focus on optimizing *Leishmania* DNA detection in sand fly.

### Methodology/Principal findings

EQA samples consisted of *Leishmania infantum* and *L. major* species, at concentrations ranging from $10^1$ to $10^4$ parasites/mL. All but one center detected all concentrations,

**Data Availability Statement:** The authors confirm that all data underlying the findings are fully available without restriction. All relevant data are within the paper and its Supporting Information files.

**Funding:** This study is co-funded by the European Commission grant 101057690, UKRI grants 10038150 and 10039289 (J.P. to F.R.-G.) and by CLIMOS project. This work also received financial support from INSERM for publication fees. The funders had no role in the study design, data collection and analysis, decision to publish, or preparation of the manuscript.

**Competing interests:** The authors have declared that no competing interests exist.

demonstrating strong diagnostic proficiency. The ability to detect low concentrations highlighted the robustness of the qPCR assay used, though variations in Cq values indicated differences in sensitivity related to technical capabilities or DNA extraction kit performance. A comparative analysis of seven DNA extraction methods identified the EZ1 DSP Virus Kit and QIAamp DNA mini-kit as the most efficient, supporting their use in standardized protocols. The study also assessed the effects of lyophilization and shipment conditions, showing no significant compromise in *Leishmania* detection despite slight variations in Cq values. Experimentally infected sand flies were included to simulate field conditions, and all centers successfully detected positive samples with varying Cq values, probably reflecting differences in infection load.

## Conclusion and significance

This study emphasizes the importance of standardized DNA extraction protocols and continuous quality assurance for accurate *Leishmania* DNA detection. The results highlight the superior performance of certain extraction kits and the need for ongoing technical training, essential for reliable leishmaniasis surveillance, particularly in field settings with low infection densities.

### Author summary

Leishmaniasis is a disease caused by *Leishmania* parasites, transmitted by sand flies, and poses a major health risk in the Mediterranean region. Monitoring the spread of infected sand flies is crucial for controlling the disease. This study focused on improving the methods used to detect *Leishmania* in sand flies by comparing different DNA extraction techniques and assessing the accuracy of these methods across nine reference laboratories. All centers, except one, efficiently detected all *Leishmania* concentrations, demonstrating proficiency in diagnostic protocols. Moreover, we found that two specific DNA extraction kits, the EZ1 DSP Virus Kit and QIAamp DNA mini-kit, were the most effective for *Leishmania* detection. We also tested how sample preparation and shipping conditions affected the results, ensuring that our methods would work in real-world settings. Even under these conditions, the detection methods proved reliable. This work helps to standardize the detection of *Leishmania*, making surveillance more accurate and consistent. Continuous training and calibration are essential to ensure uniform diagnostic performance across laboratories, enhancing epidemiological surveillance and disease control strategies and enabling appropriate treatment.

## Introduction

Leishmaniasis is a disease caused by parasites of the genus *Leishmania* and transmitted by the bite of Phlebotomine sand flies. This parasitic infection is endemic in territories around the Mediterranean basin, where it represents a significant public health concern [1]. Leishmaniasis clinical manifestations are diverse, ranging from cutaneous lesions, which may cause disfiguring ulcers on exposed parts of the body, to visceral disease with infiltration of the lymph nodes, spleen, liver, and bone marrow, causing pancytopenia and being fatal if untreated [2,3]. Despite the significant health impact of leishmaniasis worldwide, there is no substantial evidence indicating a rise in the incidence of autochthonous human cases in Europe. However,

the disease remains often underreported, leading to a possible underestimation of its true burden [4]. Hypothesis confirmed by a notable increase of autochthonous canine leishmaniasis cases [5]. Furthermore, the geographical distribution of leishmaniasis is changing. New foci of infection are emerging in areas previously considered non-endemic, while old foci are re-emerging [5–7]. Triggering factors contributing to this evolving scenario include climate change, which affects the distribution and behavior of sand fly vectors, increased movement of people and animals and trade activities, which can cause parasite introduction into new areas. Overall, the dynamic epidemiology of leishmaniasis in Europe underscores the need for vigilant surveillance and reporting systems.

Effective epidemiologic surveillance of *Leishmania*-infected sand flies could be considered an essential tool for understanding and controlling the spread of leishmaniasis. To achieve this goal, standardized protocols are required to accurately compare the distribution areas and the prevalence of sand fly infection. Real-time quantitative PCR (qPCR), which is a sensitive and specific method for detecting *Leishmania* DNA, plays a crucial role in this process [8]. Utilizing consistent amplification conditions and similar extraction protocols across different laboratories is vital for ensuring that the results are comparable over time and across various geographical areas [9]. Moreover, reliable data merged from different regions can help in mapping the spread of the disease and in understanding the factors driving its transmission, such as climate change, urbanization, and movements of infected hosts and vectors [10].

Despite the importance of these techniques, there is currently a lack of evaluation in nucleic acid extraction and qPCR techniques for *Leishmania* diagnosis. This deficiency represents a significant gap in the epidemiologic surveillance framework. Without standardized and validated methods, the reliability of data collected from different studies can be compromised, making it difficult to draw accurate comparisons and conclusions. Laboratory efficiency has been compared throughout European countries for the diagnosis of other parasitic or fungal diseases, such as toxoplasmosis [11,12], histoplasmosis [9] and *Pneumocystis* pneumonia [13]. Regarding leishmaniasis, a European study [14] has previously compared the accuracy of species identification by molecular methods, but no such initiative has been implemented for evaluating *Leishmania* detection by qPCR. The sensitivity of a qPCR method is highly dependent on the extraction method used, as shown in several studies conducted by the French National Reference Center for toxoplasmosis [15,16], but no study evaluated DNA extraction protocols for *Leishmania* parasites. As for amplification method, the kinetoplastid DNA (kDNA) target [17] is widely recognized as a very sensitive qPCR target [18] due to a high copy number of kDNA in the parasite mitochondria for Old World *Leishmania* species and is used in many studies around the world [19–21].

In this context, the objectives of this study were twofold: (i) to analyze the performance of various DNA extraction protocols for detecting *L. infantum* and *L. major*, by qPCR and (ii) to conduct an External Quality Assessment (EQA) aimed at evaluating the effectiveness and consistency of *Leishmania* DNA detection methods across different laboratories. This multicenter study involved nine reference laboratories, participating in the European project CLIMOS (http://www.climos-project.eu) which collects data on sand fly infection, and aimed at ensuring the reliability and comparability of *Leishmania* detection methods from sand flies across different regions and laboratories, thereby enhancing the accuracy of epidemiological surveillance and contributing to more effective disease control strategies.

## Methods

### Participants and study design

The Laboratory of Parasitology of Rennes University/Institut National de la Santé et de la Recherche Médicale (INSERM) (Rennes, France), which is a reference laboratory for the

diagnosis of leishmaniasis and other parasitic and fungal infections, was in charge of developing standard operation procedures (SOP) for *Leishmania* extraction from sand flies and oversaw the implementation of the EQA program for CLIMOS. INSERM prepared the EQA samples and evaluated the various extraction methods used by eight European and non-European laboratories involved in the project, located in 6 countries, including: the reference center (INSERM, Rennes, France), Ege University (EGE, Izmir, Turkey), Hacettepe Universitesi (HACETTEPE, Ankara, Turkey), Institute of Hygiene and Tropical Medicine, Universidade Nova de Lisboa (UNL, Lisboa, Portugal), Jerusalem Public Health laboratories, Ministry of Health (IMOH, Jerusalem, Israel), Istituto Superiore di Sanita (ISS, Roma, Italy), Medizinische Universitaet Wien (MEDUNI VIENNA, Wien, Austria), Turkiye Cumhuriyeti Saglik Bakanligi (MOH, Ankara, Turkey) and Universidad de Murcia (UM, Murcia, Spain). The participating centers other than the reference center were designated as "Center 1 to Center 8". The EQA program for *Leishmania* DNA extraction and qPCR analysis involved testing cultured parasites and experimentally infected sand flies.

## Sand fly samples

For all experiments and EQA samples, we used *Phlebotomus perniciosus* from well-adapted laboratory colonies. Sand flies were provided by Istituto Superiore di Sanita (ISS) (Roma, Italy) and Hacettepe Universitesi (HU) (Ankara, Turkey) for uninfected specimens, and by Charles University (CUNI) (Prague, Czech Republic) for experimentally infected ones [22].

## *Leishmania* species and preparation of EQA samples

Two species of *Leishmania* were used, *L. infantum* #REN-12-02 and *L. major* #REN-22-02 (both cryopreserved at the Biological Resource Center of the Rennes University Hospital and Leishmaniasis Reference Center of Montpellier University Hospital) for the comparison of DNA extraction techniques, preparation of EQA samples and EQA validation. Promastigotes were maintained in an incubator at 26˚C by weekly transfers in T25 flask containing M199 medium (Sigma) supplemented with 10% inactivated fetal calf serum, 1% HEPES, 1% Penicillin-Streptomycin, 1% hypoxanthin, 0.2% hemin, 0.1% biotin and 0.4% biopterin.

Five serial 1:10 dilutions of each species containing $10^5$, $10^4$, $10^3$, $10^2$ and $10^1$ parasite/mL were prepared, starting with, 1 mL of homogenized broth culture. Dilutions were carried out in a 5% formalinized Dulbecco's phosphate-buffered saline (DPBS) solution. Promastigotes were counted using the standardized KOVA cell chamber system, according to the protocol established by the supplier (Kova International, California, USA). To ensure accuracy, the counting was realized in 3 cells and by two different operators. Ready to use, parasite suspensions were aliquoted into 1.5 mL tubes and directly stored at -20˚C or lyophilized and stored at -20˚C until use. A set of lyophilized samples was kept at room temperature for 3 weeks to evaluate the impact of storage conditions on qPCR results.

## EQA sample processing

All centers received a panel of ten EQA samples, consisting of eight lyophilized (*i.e. L. major* and *L. infantum* at $10^4$, $10^3$, $10^2$ and $10^1$ parasite/mL) and two liquid samples (*i.e.* one uninfected and one experimentally infected sand fly in 200 μL of phosphate buffered saline (PBS)). At reception, samples were stored at −20˚C until further testing. Lyophilized samples were rehydrated with 200 μL of PCR-quality water and sand fly samples were processed like any sand fly collected from the field for analysis, *i.e.* ground in a final volume of 700 μL of PBS and incubated at 56˚C during 2 hours with proteinase K. Then, extractions were realized with an

**Table 1. Methods used for *Leishmania* nucleic acids extraction and detection in the different center.**

| Center | | Extraction kit | Extraction device | Elution volume | Amplification device |
|---|---|---|---|---|---|
| Reference center | INSERM | EZ1 DSP Virus Kit | EZ1 extraction device | 90 | StepOne Real-time PCR System |
| 1 | ISS | RSC Blood DNA | Maxwell RSC 16 instrument | 50 | Biorad iQ5 |
| 2 | MEDUNI VIENNA | Allprep DNA/RNA micro kit | Manual extraction | 90 | Biorad CFX96 Real-time system |
| 3 | UNL | QIAmp viral RNA mini kit | Manual extraction | 90 | Rotor Gene 3000 |
| 4 | UM | Kit 1: RSC Viral TNA | Maxwell RSC 16 instrument | 90 | QuantStudio 5 Real-Time PCR |
| | | Kit 2: RSC Blood DNA | | 50 | |
| 5 | HU | Allprep DNA/RNA mini kit | Manual extraction | 90 | StepOnePlus Real-time PCR System |
| 6 | MOH | EZ1&2 virus mini kit | EZ1 extraction device | 90 | Biorad CFX96 Real-time system |
| 7 | EGE | DNeasy Blood & Tissue Kit | Manual extraction | 90 | Rotor-Gene Q |
| 8 | IMOH | Mag-Bind Blood & Tissue DNA Kit | Manual extraction | 90 | Biorad CFX96 Real-time system |

extraction volume of 400 µL (200 µL of EQA sample and 200 µL lysis buffer) and an elution volume between 50 and 90 µL according to each center technique and equipment (Table 1).

All partners employed the same qPCR method [17] based on the amplification of a kinetoplast DNA (kDNA) minicircle sequence with primers and Taqman probe: 5'-CTT-TTC-TGG-TCC-TCC-GGG-TAGG, 5'-CCA-CCC-GGC-CCT-ATT-TTA-CAC-CAA and 5' FAM-TTT-TCG-CAG-AAC-GCC-CCT-ACC-CGC-3' TAMRA, respectively, provided by the reference center. Each 25 µL qPCR reaction mix included 5 µL of DNA sample, 12.5 µL of TaqMan Universal Master Mix 2X and a final concentration of 0.5 µM of primers and 0.2 µM of probe. DNA was amplified using the following conditions: initial step at 95˚C for 10 min, followed by 45 cycles of 15 sec at 95˚C and 1 min at 60˚C. Participating centers used their own qPCR device (Table 1), realized the amplification in triplicates and included their own positive and negative controls. The qPCR quantification cycle (Cq) defined as the cycle at which near logarithmic product amplification takes place, was used as a semi-quantitative measure of parasite DNA concentration [23].

## Comparison of DNA extraction techniques

As the amplification method was the same for all participating centers, we suspected that variations might appear related, at least partly, to the extraction method used. Therefore, we undertook the evaluation of seven extraction methods, including some used by the participating centers (Table 1), and additional ones which were designed to purify total nucleic acids and could offer the opportunity to detect simultaneously *Phlebovirus*, also transmitted by sand flies.

Seven kits were compared for *Leishmania* DNA extraction, including the following five manual extraction kits: EZ1 DSP Virus Kit using EZ1 extraction device (Qiagen, Hilden, Germany), RNeasy mini kit (Qiagen, Hilden, Germany), QIAamp DNA mini kit (Qiagen, Hilden, Germany), Allprep DNA/RNA mini kit (Qiagen, Hilden, Germany), QIAamp viral RNA mini kit (Qiagen, Hilden, Germany) and two automated extraction kits: RSC Viral TNA (Promega, Southampton, England) and RSC Blood DNA (Promega, Southampton, England) using Maxwell RSC 48 instrument (Promega). Amplifications were realized using a QuantStudio 5 Real-Time PCR System (Applied Biosystems, Thermo Fisher Scientific, Villebon-sur-Yvette, France). Liquid samples, containing parasite suspensions, aliquoted in small vials and stored at -20˚C were used for this evaluation, to avoid possible variations due to the lyophilization process and reconstitution. Extractions were performed in triplicate from 3 independent vials of each concentration, according to the manufacturer's instructions. Amplifications were also performed in triplicates, using 5 µL of DNA in a final volume of 25 µL as described above.

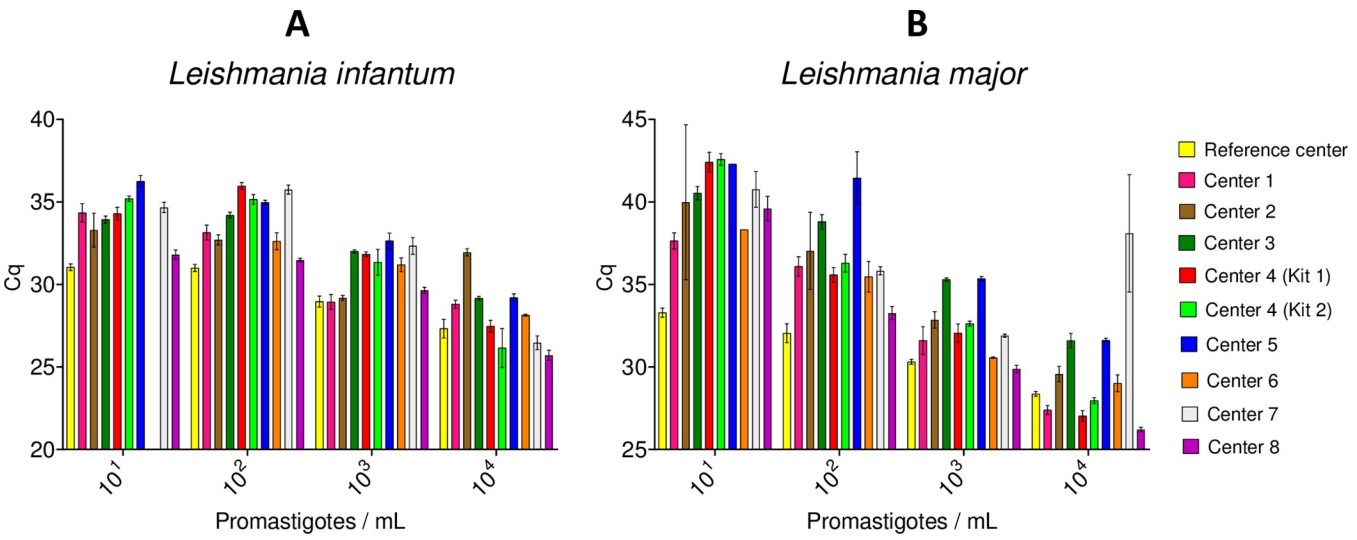

**Fig 1.** PCR results for EQA samples of *Leishmania infantum* (A) and *Leishmania major* (B) promastigotes at indicated concentrations by participating centers (mean Cq ± SD of triplicate amplification).

## Assessment of external conditions potentially influencing individual performances

First, the variability of *Leishmania* spp. DNA detection of infected sand flies using seven individuals extracted with the same kit was assessed (EZ1 DSP Virus Kit using EZ1). Then, the process (*i.e.* lyophilization and shipment conditions) was tested through three experiments. The impact of lyophilization was assessed by DNA extraction of *L. infantum* and *L. major* aliquots at four concentrations ($10^4$, $10^3$, $10^2$ and $10^1$), before and after lyophilization. Second, to ensure there was no impact of shipment conditions on sample quality, results obtained with samples stored at room temperature (RT) for 3 weeks and samples stored at –20˚C for the same time, before DNA extraction and amplification were compared. Third, the potential inhibitory effect of sand fly DNA on the detection of low amounts of *Leishmania* DNA was tested. For this purpose, pools of sand flies (30 individuals, 15 males and 15 females) were spiked with 100 or 1000 *Leishmania* (*L. major* or *L. infantum*) promastigotes and ground in a final volume of 700 µL of PBS, mimicking usual practice for field studies. The same numbers of *Leishmania* without sand flies were used as controls. Homogenates were submitted to a 2-hour heating step with proteinase K at 56˚C before DNA extraction. Two hundred µL were used for DNA extraction using EZ1 DSP Virus Kit (Qiagen, Hilden, Germany) according to the manufacturer's instructions and eluted in 90 µL of elution buffer.

Amplification was carried out using a QuantStudio 5 Real-Time PCR device (Applied Biosystems). Extractions and amplifications were performed in triplicates.

## Statistical analysis

Results were presented as mean ±SD of quantification cycle (Cq) values of amplification of each parasite concentration for each center. They were compared using two-way ANOVA or mixed-effects analyses (if missing data were present) and a Tukey's multiple comparisons test as post-hoc analyses. All analyses and graphics were realized with GraphPad Prism Software version 9.

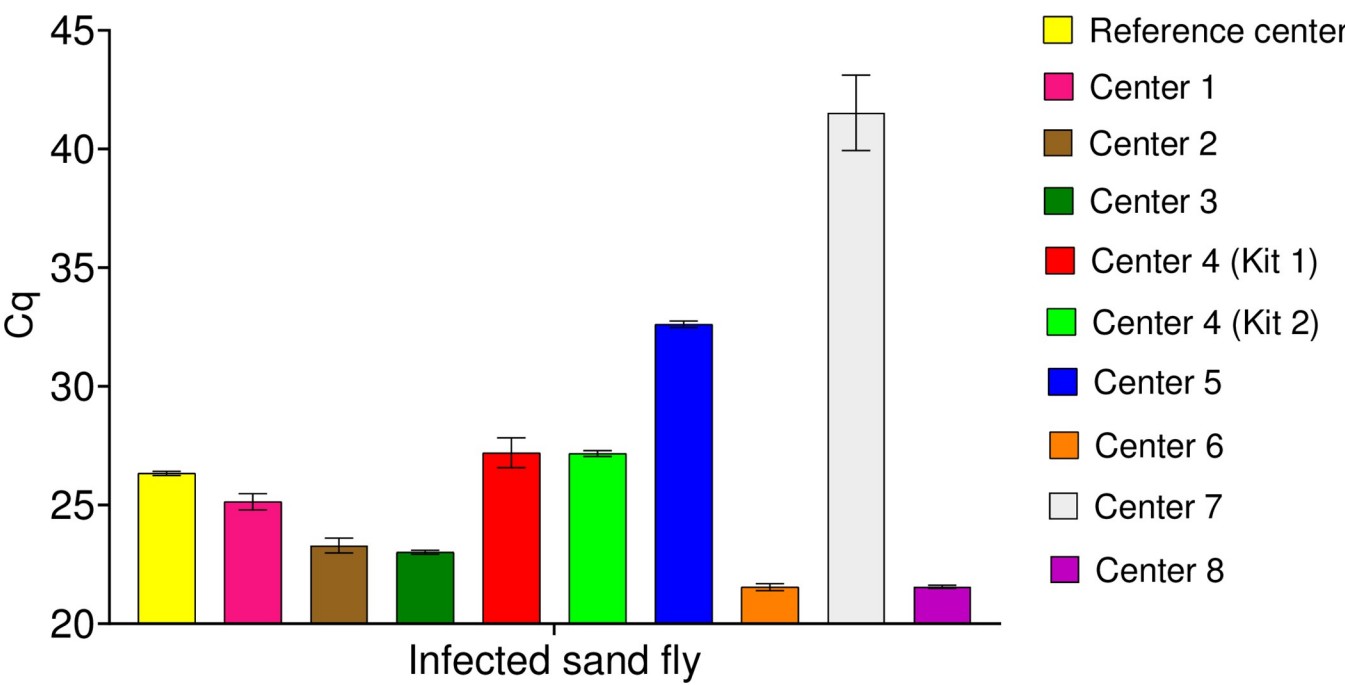

**Fig 2. Detection of *Leishmania infantum* DNA from an infected sand fly.** Real-time quantitative PCR results of participating centers (mean Cq ± SD of triplicate amplification).

## Results

### Multicenter qPCR analysis of EQA samples

Results of DNA amplification by qPCR of the two *Leishmania* species at four different concentrations, obtained by the eight centers, are provided in Fig 1. All centers correctly amplified positive samples except center 6 which failed to amplify *L. infantum* at $10^1$/mL concentration. For *L. infantum*, mean Cq values across centers ranged from approximately 26 to 32, 29 to 32 and 31 to 36 for the $10^4$/mL, $10^3$/mL and $10^2$/mL parasite concentrations, respectively (Fig 1A). Similarly, the mean Cq values obtained for *L. major* showed notable inter-center variations, with overall higher mean Cq values compared to those obtained in the *L. infantum* assay (Fig 1B). The mean Cq values ranged from 26 to 38 for samples with $10^4$/mL parasites and from approximately 30 to 45 for samples with $10^2$/mL parasites. At the lowest *L. major* concentration of $10^1$/mL parasites, Cq values ranged from 33 to 43. All participating centers also accurately detected the samples containing a *L. infantum* experimentally infected sand fly, with variable mean Cq ranging from 22 to 42 (Fig 2).

### Comparison of extraction protocols

For both *L. infantum* and *L. major*, the use of EZ1 DSP Virus Kit, QIAamp DNA mini kit, Allprep DNA/RNA mini kit, and RSC Blood DNA yielded the lowest Cq values for all concentrations, indicating they are the most efficient kits in extracting DNA from *Leishmania*. Even though these four kits were associated with low Cq values, extraction with EZ1 DSP Virus Kit offered the best overall efficiency since mean Cq values were significantly lower than those for other kits at most *L. infantum* and *L. major* concentrations (Tables 2 and 3). Conversely, RNeasy mini kit, QIAamp viral RNA mini kit, and RSC Viral TNA showed higher Cq values, indicating lower efficiency in extracting DNA. The detailed comparisons between the various

**Table 2. P-value obtained with Tukey's multiple comparisons test between mean Cq values of *Leishmania infantum* kDNA amplification after DNA extraction with indicated kits.**

| Kit name | Kit compared | Concentration | | | | |
|---|---|---|---|---|---|---|
| | | $10^1$ | $10^2$ | $10^3$ | $10^4$ | $10^5$ |
| EZ1 DSP Virus Kit | vs RSC Blood DNA | 0.0003 | <0.0001 | <0.0001 | <0.0001 | <0.0001 |
| | vs QIAamp viral RNA mini kit | 0.5302 | <0.0001 | <0.0001 | <0.0001 | <0.0001 |
| | vs RNeasy mini kit | 0.0014 | <0.0001 | <0.0001 | <0.0001 | 0.0001 |
| | vs RSC Viral TNA | NA | 0.0275 | <0.0001 | <0.0001 | <0.0001 |
| | vs Allprep DNA/RNA mini kit | 0.0009 | <0.0001 | <0.0001 | <0.0001 | <0.0001 |
| | vs QIAamp DNA mini kit | 0.5452 | <0.0001 | 0.0001 | <0.0001 | <0.0001 |
| RSC Blood DNA | vs QIAamp viral RNA mini kit | 0.9680 | 0.0188 | 0.0086 | 0.0067 | <0.0001 |
| | vs RNeasy mini kit | 0.1585 | 0.2685 | 0.2256 | 0.0366 | 0.0028 |
| | vs RSC Viral TNA | NA | 0.5402 | <0.0001 | <0.0001 | 0.0048 |
| | vs Allprep DNA/RNA mini kit | 0.3833 | 0.1166 | 0.6613 | 0.7066 | <0.0001 |
| | vs QIAamp DNA mini kit | 0.0019 | <0.0001 | 0.0167 | 0.0030 | 0.2812 |
| QIAamp viral RNA mini kit | vs RNeasy mini kit | >0.9999 | 0.6866 | 0.6467 | 0.9988 | 0.0754 |
| | vs RSC Viral TNA | NA | >0.9999 | <0.0001 | 0.1162 | <0.0001 |
| | vs Allprep DNA/RNA mini kit | 0.8537 | 0.0016 | 0.0016 | 0.0011 | 0.8189 |
| | vs QIAamp DNA mini kit | 0.6189 | 0.0002 | 0.0002 | <0.0001 | <0.0001 |
| RNeasy mini kit | vs RSC Viral TNA | NA | 0.9847 | <0.0001 | 0.0671 | 0.0016 |
| | vs Allprep DNA/RNA mini kit | 0.0317 | 0.0273 | 0.0214 | 0.0064 | 0.1029 |
| | vs QIAamp DNA mini kit | 0.0054 | 0.0012 | 0.0015 | 0.0004 | 0.0049 |
| RSC Viral TNA | vs Allprep DNA/RNA mini kit | NA | 0.2586 | <0.0001 | <0.0001 | <0.0001 |
| | vs QIAamp DNA mini kit | NA | 0.0941 | <0.0001 | <0.0001 | 0.0106 |
| Allprep DNA/RNA mini kit | vs QIAamp DNA mini kit | 0.0010 | 0.0525 | <0.0001 | 0.0002 | <0.0001 |

NA: Not available due to an excessive amount of missing data for some kits at low concentrations.

extraction kits for *L. infantum* are depicted in Fig 3A and p-values are summarized in Table 2, and those for *L. major* are illustrated in Fig 3B and summarized in Table 3.

## Assessment of external conditions

The variability of infection levels in seven sand flies experimentally infected with *L. infantum* is presented in Fig 4. The data presented highlight significant differences in infection intensity among infected individuals, with mean Cq values ranging from 17 to 38, for sand fly DNA extracts obtained using the same assay (EZ1 DSP Virus Kit).

The impact of lyophilization, shipment conditions and presence of sand fly DNA in mean Cq values of EQA samples is depicted in Fig 5A, 5B and 5C, respectively. Lyophilization showed no impact on mean Cq values for low parasite concentrations ($10^1$ and $10^2$ for *L. infantum*, $10^1$ for *L. major*). Instead, Cq were significantly greater for higher parasites concentrations ($10^3$ for *L. infantum*, $10^2$ and $10^3$ for *L. major*) (Fig 5A). No influences of the storage conditions were noticed, as preservation at -20°C compared to room temperature showed no significant differences in mean Cq values (Fig 5B). Moreover, the presence of DNA from 30 sand flies did not affect *Leishmania* spp. detection at low concentrations. In fact, the efficiency of *L. major* DNA amplification was even better in presence of sand fly DNA (lower Cq values, p-<0.05) (Fig 5C).

## Discussion

The implementation of reliable techniques is crucial when they form the core of pathogen surveillance programs, comparing endemicity levels between countries. In this context, it was

**Table 3. P-value obtained with Tukey's multiple comparisons test between mean Cq values of *Leishmania* major kDNA amplification after DNA extraction with indicated kits.**

| Kit name | Kit compared | Concentration | | | | |
|---|---|---|---|---|---|---|
| | | $10^1$ | $10^2$ | $10^3$ | $10^4$ | $10^5$ |
| EZ1 DSP Virus Kit | vs RSC Blood DNA | <0.0001 | 0.0924 | <0.0001 | <0.0001 | <0.0001 |
| | vs QIAamp viral RNA mini kit | <0.0001 | 0.0084 | <0.0001 | <0.0001 | <0.0001 |
| | vs RNeasy mini kit | <0.0001 | 0.0004 | <0.0001 | <0.0001 | <0.0001 |
| | vs RSC Viral TNA | 0.4009 | <0.0001 | <0.0001 | <0.0001 | <0.0001 |
| | vs Allprep DNA/RNA mini kit | <0.0001 | 0.0111 | <0.0001 | <0.0001 | 0.0001 |
| | vs QIAamp DNA mini kit | 0.0001 | 0.4043 | <0.0001 | <0.0001 | <0.0001 |
| RSC Blood DNA | vs QIAamp viral RNA mini kit | 0.9878 | 0.2849 | 0.9720 | 0.1075 | <0.0001 |
| | vs RNeasy mini kit | 0.9882 | 0.0010 | 0.0033 | 0.0001 | <0.0001 |
| | vs RSC Viral TNA | >0.9999 | 0.0114 | 0.0155 | <0.0001 | 0.8399 |
| | vs Allprep DNA/RNA mini kit | >0.9999 | 0.1010 | 0.9988 | 0.0246 | 0.0091 |
| | vs QIAamp DNA mini kit | 0.2100 | 0.1149 | 0.8099 | 0.3685 | 0.0043 |
| QIAamp viral RNA mini kit | vs RNeasy mini kit | 0.6184 | 0.6500 | 0.1607 | 0.0018 | 0.0001 |
| | vs RSC Viral TNA | >0.9999 | 0.0055 | 0.0294 | <0.0001 | <0.0001 |
| | vs Allprep DNA/RNA mini kit | 0.9581 | 0.9851 | 0.9197 | 0.9973 | 0.4575 |
| | vs QIAamp DNA mini kit | 0.0512 | 0.0352 | 0.9998 | 0.0080 | 0.0003 |
| RNeasy mini kit | vs RSC Viral TNA | 0.9973 | 0.0302 | 0.2577 | 0.8465 | 0.0001 |
| | vs Allprep DNA/RNA mini kit | 0.9992 | 0.0445 | 0.0020 | 0.0018 | 0.0464 |
| | vs QIAamp DNA mini kit | 0.0643 | 0.0001 | 0.0099 | 0.0001 | 0.0001 |
| RSC Viral TNA | vs Allprep DNA/RNA mini kit | >0.9999 | 0.0236 | 0.0131 | <0.0001 | 0.0114 |
| | vs QIAamp DNA mini kit | 0.9407 | 0.0507 | 0.0223 | <0.0001 | 0.0005 |
| Allprep NA/RNA mini kit | vs QIAamp DNA mini kit | 0.2013 | 0.0002 | 0.5560 | <0.0001 | 0.0210 |

pertinent to implement standard operating procedures to ensure high performance among research centers involved in sand fly and SFBDs surveillance, using the same qPCR amplification method [17] and amplification conditions, to normalize the interpretation of results. This qPCR targets kinetoplastid DNA sequence which is highly repeated, thus enhances the sensitivity of detection. For this reason, together with its excellent specificity [18], this qPCR is

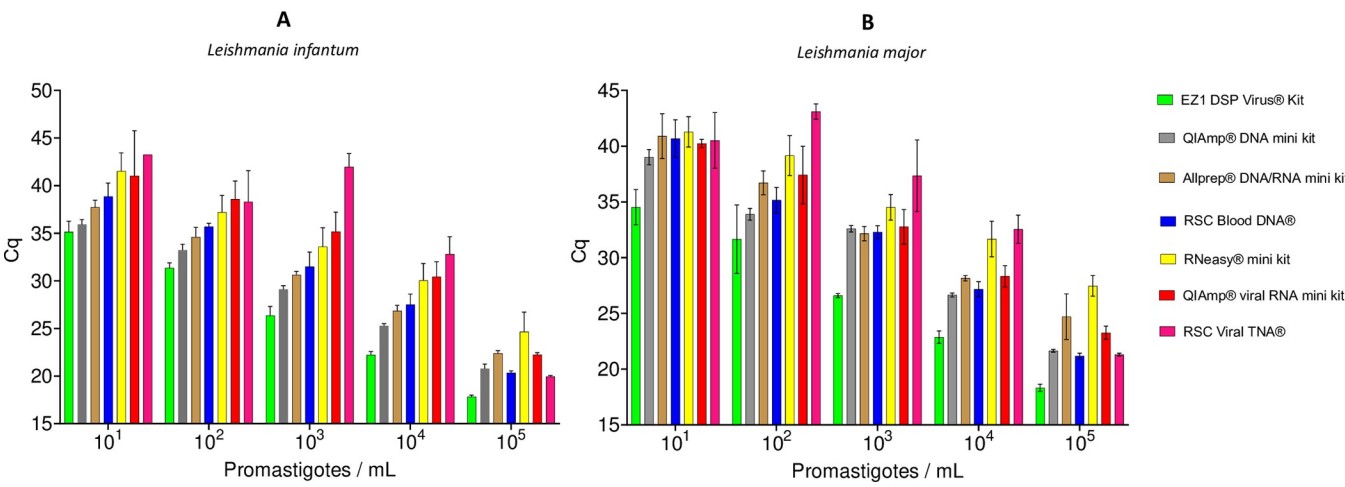

**Fig 3.** Efficiency of *Leishmania infantum* (A) and *Leishmania major* (B) detection following nucleic acids extraction using various kits. Results show mean Cq ± SD of triplicate amplification for each parasite concentration.

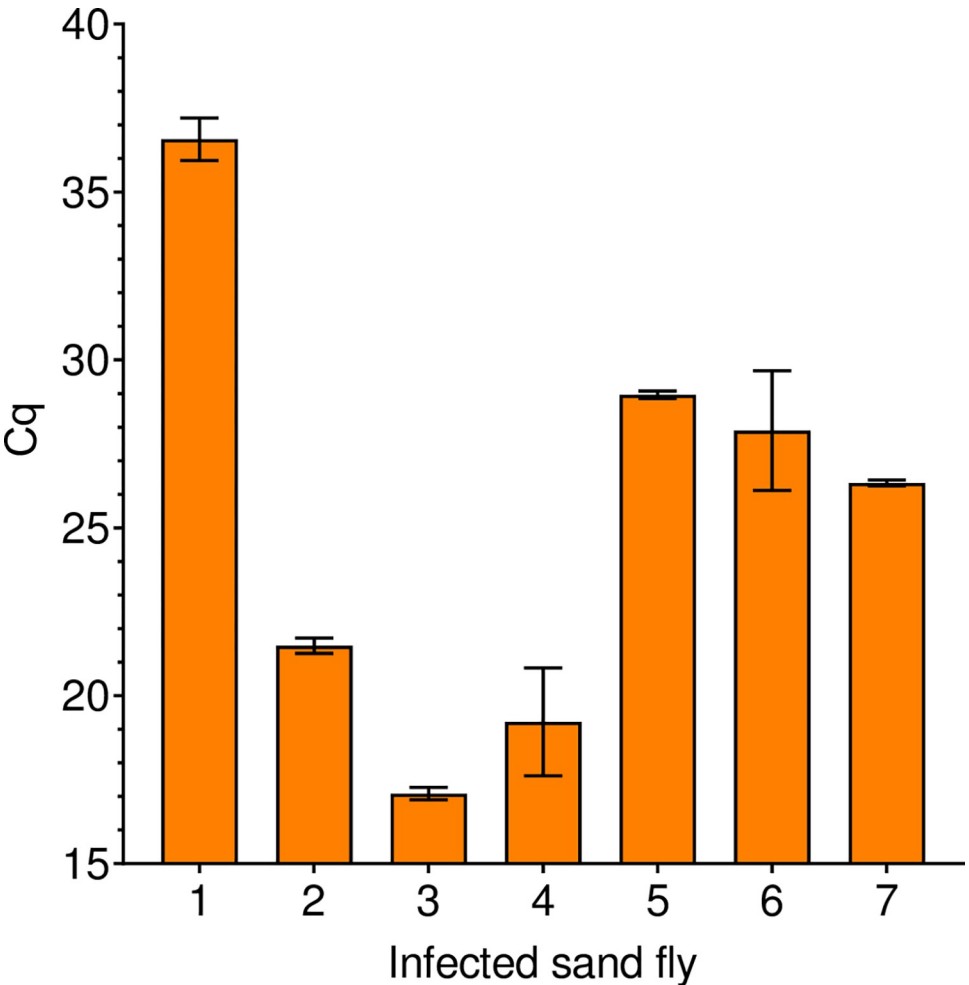

**Fig 4. Variability of infection levels in sand flies infected with *Leishmania infantum* under laboratory conditions.** Results show mean Cq ± SD of triplicate amplification for each individual sand fly (7 specimens).

widely used for the diagnosis of leishmaniasis in humans [19,20]. However, depending on their own equipment and facilities, the project partners used different extraction methods and qPCR devices. Thus, it was necessary to confirm that centers had comparable performances. With this aim, the Laboratory of Parasitology of Rennes, highly reputed center for evaluation of molecular techniques in the field of human diagnosis [24–26], was in charge of the implementation of an external quality assessment program.

Overall, all centers but one reliably detected all EQA samples corresponding to *L. infantum* and *L. major* concentrations ranging from $10^1$ to $10^4$/mL. The error types (ET) in Figs 1 and 2 are low for most datasets and reflect a highly repeatable amplification procedure in most centers. Higher ET are however expected for low concentrations due to Poisson's law, particularly in the presence of PCR inhibitors. The center who did not detect *L. infantum* ($10^1$/mL) detected *L. major* ($10^1$/mL) only once. Lower sensitivity in qPCR amplification detection could be related to lower performance of the DNA extraction method or of the qPCR master mix used. Unfortunately, it was not possible to retest the $10^1$/mL samples due to lack of remaining DNA eluate. Notwithstanding this, this laboratory was able to amplify the sample containing $10^1$/mL *L. major* and all other samples with higher parasite concentrations. The

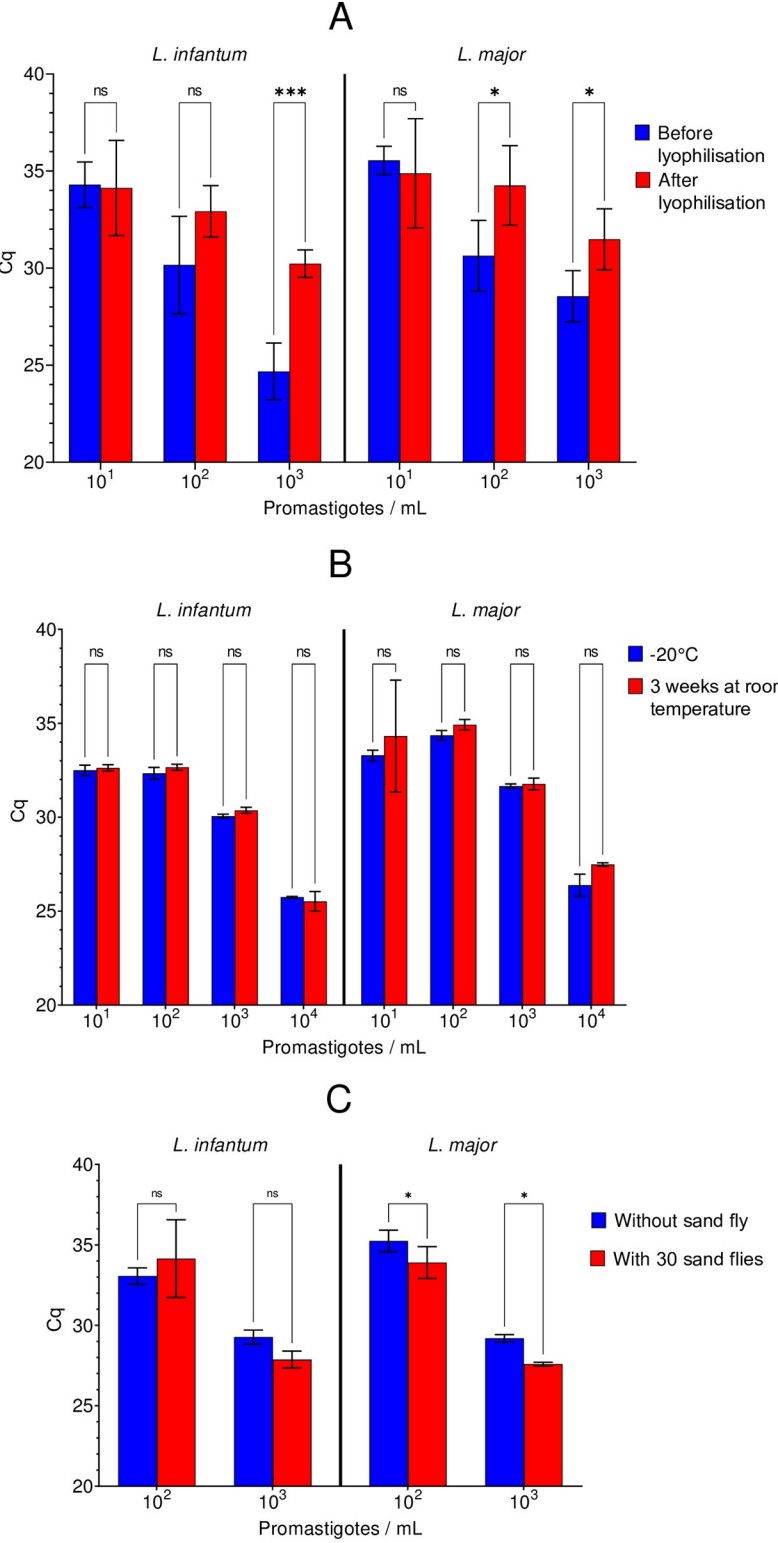

**Fig 5.** Impact of lyophilization (A), temperature of storage (B), sand fly DNA inhibitors (C) on the efficiency of *Leishmania* DNA amplification. ns: not significant; * p<0.05; ** p<0.01; *** p<0.001.

implications of failing to detect low parasite concentrations are probably low, given that most infected sand flies are likely to contain large parasite numbers, as demonstrated in the present study. The ability of all centers to detect low concentrations of *Leishmania* demonstrates that all partners are duly trained in performing surveillance of *Leishmania*-infected sand flies. Standardization of these methods enables to track changes in infection rates accurately and identify emerging hotspots of transmission. For instance, consistent use of qPCR allows the detection of even low levels of parasite DNA in sand flies, which is critical for early warning and timely intervention in areas where leishmaniasis is spreading. However, the variability in the EQA Cq values provides critical insights into the performance and sensitivity of the diagnostic assay employed. The observed differences in mean Cq values could be attributable to the technical proficiency of the operator, the differences between equipment and positive threshold setting, the DNA extraction kit used, or sample-related factors. The EQA process was evaluated and validated before shipment to ensure reliable comparison of laboratory performances. The variations in DNA yield post-lyophilization could partially explain some discrepancies between centers but would hardly explain the high range of Cq observed for a same concentration. Noteworthy, no significant impact of shipment conditions was observed, thus sample degradation was unlikely to be responsible for the lower sensitivity observed for some laboratories. This finding highlights the need for ongoing training and standardization to ensure uniform diagnostic performance across different laboratories.

The overall higher Cq values observed for *L. major* compared to *L. infantum* could be explained by several factors. One key aspect involves the minicircle kinetoplast DNA (kDNA), which is commonly targeted in qPCR assays for *Leishmania* detection. The copy number of kDNA can vary significantly between species within the *Leishmania* genus and even between isolates of the same species [27]. It is possible that *L. infantum* has a higher kDNA copy number than *L. major*, resulting in lower Cq values for *L. infantum*. Additionally, the efficiency of the qPCR assay may differ between species due to variations in the target DNA sequences.

All centers were able to detect *Leishmania* DNA in the sample containing a sand fly experimentally infected with *L. infantum*, although differing in quantification cycle (Cq) values. This difference underscores the heterogeneity in host-pathogen interactions at the individual level, even under standardized infection conditions. Indeed, this variability could be the result of fluctuations which are expected and deemed normal within the context of experimental infection, influenced by factors such as the size of the blood meal and the age of the sand fly, among others [28]. Despite these variations, the effectiveness of all centers in detecting positive samples was evident, showcasing their proficiency in handling the diagnostic protocols.

After ruling out problems of curve interpretation and Cq threshold variations, extraction methods were evaluated, as a potential source of variability. Results suggest that the choice of the extraction kit may markedly influence the sensitivity of *Leishmania* DNA detection. Taken together, these results may help to understand the discrepancies in Cq values observed between centers. It was observed that the EZ1 DSP Virus Kit and QIAamp DNA mini kit had the best performances for both *Leishmania* species amplification, independently of the parasite concentration. Implementing these kits across laboratories could standardize and improve the consistency and reliability of *Leishmania*-infected sand fly detection. As expected, the use of automated extraction systems, such as the EZ1 robot or Maxwell device, led to reduced variability, as shown by low error types of triplicate extractions. Despite the superior performance of some DNA extraction kits, it is important to insist that all of them yielded suitable DNA template for effective qPCR detection of *Leishmania*-infected sand flies. However, there remains room for improvement in analyzing low concentrations to ensure high diagnostic validity in *Leishmania* spp. surveillance programs.

As might be expected, the kits developed specifically for DNA extractions (such as EZ1 DSP Virus Kit, QIAamp DNA mini kit, Allprep DNA/RNA mini kit, and RSC Blood DNA) outperformed those designed for RNA-extraction (such as RNeasy mini kit and QIAamp viral RNA mini kit), with avoidance of the DNAase digestion step. These finding implies that field studies aiming at monitoring sand fly-borne infections, *i.e.* *Leishmania* and phleboviruses, should use a total nucleic acid extraction kit (*e.g.* Allprep DNA/RNA mini kit) for both pathogens, or use two extraction kits designed for DNA and RNA purification, respectively.

To conclude, all participating centers were proficient in carrying out the diagnostic protocols in the EQA. The detailed comparisons and analyses of different extraction kits for *Leishmania* underscore the importance of selecting the appropriate protocol to ensure high-quality DNA amplification. The benefit of automated extraction, support their adoption across laboratories. While all tested kits are effective, optimizing protocols for low concentration samples remains a key area for improvement to enhance the exhaustive and reliable detection of *Leishmania* in field studies. These results emphasize the importance of standardized protocols and continuous quality assurance to maintain high diagnostic accuracy, which is essential for effective leishmaniasis surveillance in field settings where low concentrations of infection are common.

## Supporting information

**S1 Table. Raw qPCR data, including *Leishmania* species, concentration, experimental conditions, and replicate information for all samples analyzed in the study.**
(XLSX)

## Acknowledgments

The authors thank Nazli Ayhan, Rémi Charrel and Laurence Thirion from the UVE laboratory (Unité des Virus Emergents, Marseille, France) for EQA vials lyophilization. They also thank Jovana Sadlova and Petr Volf from Charles University (Department of Parasitology, Prague, Czech Republic) for providing infected sand flies and Jean-Pierre Gangneux (Université de Rennes, Rennes, France) for providing the *Leishmania* isolates. Finally, U.B., M.M., and G.V. thank the General Directorate of Public Health/National Parasitology Reference Laboratory (Ankara, Turkey) for local support. This study is catalogued by the CLIMOS Scientific Committee as CLIMOS number 011 (www.climos-project.eu).

## Disclaimer

## Author Contributions

**Conceptualization:** Jorian Prudhomme, Florence Robert-Gangneux.

**Formal analysis:** Jorian Prudhomme, Aymeric Delabarre, Florence Robert-Gangneux.

**Investigation:** Jorian Prudhomme, Aymeric Delabarre, Bulent Alten, Umut Berberoglu, Eduardo Berriatua, Gioia Bongiorno, José Manuel Cristovao, Maya Davidovich-Cohen, Trentina Di Muccio, Ozge Erisoz Kasap, Eleonora Fiorentino, Oscar D. Kirstein, Edwin Kniha, Carla Maia, Mesut Mungan, Clara Muñoz-Hernández, Muhammed Nalçaci, Gizem Oguz Kaskan, Yusuf Ozbel, Seray Ozensoy Toz, Ricardo Parreira, Katharina Platzgummer, Ceylan Polat, José Risueño, Liora Studentsky, Gamze Varol, Julia Walochnik, Kardelen Yetişmiş.

**Methodology:** Jorian Prudhomme, Aymeric Delabarre, Florence Robert-Gangneux.

**Supervision:** Florence Robert-Gangneux.

**Writing – original draft:** Jorian Prudhomme, Aymeric Delabarre, Florence Robert-Gangneux.

**Writing – review & editing:** Jorian Prudhomme, Aymeric Delabarre, Bulent Alten, Umut Berberoglu, Eduardo Berriatua, Gioia Bongiorno, José Manuel Cristovao, Maya Davidovich-Cohen, Trentina Di Muccio, Ozge Erisoz Kasap, Eleonora Fiorentino, Oscar D. Kirstein, Edwin Kniha, Carla Maia, Mesut Mungan, Clara Muñoz-Hernández, Muhammed Nalçaci, Gizem Oguz Kaskan, Yusuf Ozbel, Seray Ozensoy Toz, Ricardo Parreira, Katharina Platzgummer, Ceylan Polat, José Risueño, Liora Studentsky, Gamze Varol, Julia Walochnik, Kardelen Yetişmiş, Florence Robert-Gangneux.

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
