## [Decision Letter · Decision Letter 0]

8 Oct 2024

Dear Dr. PRUDHOMME,

Thank you very much for submitting your manuscript "Performance evaluation of nine reference centers for effective surveillance of *Leishmania*-infected Phlebotomine sand flies and basis for technical recommendations" for consideration at PLOS Neglected Tropical Diseases. As with all papers reviewed by the journal, your manuscript was reviewed by members of the editorial board and by several independent reviewers. In light of the reviews (below this email), we would like to invite the resubmission of a significantly-revised version that takes into account the reviewers' comments. 

We cannot make any decision about publication until we have seen the revised manuscript and your response to the reviewers' comments. Your revised manuscript is also likely to be sent to reviewers for further evaluation.

Sincerely,

Clarence Mang'era, PhD

Guest Editor

Paul Mireji

Section Editor

Reviewer's Responses to Questions

**Key Review Criteria Required for Acceptance?**

**Methods**

-Are the objectives of the study clearly articulated with a clear testable hypothesis stated?

-Is the study design appropriate to address the stated objectives?

-Is the population clearly described and appropriate for the hypothesis being tested?

-Is the sample size sufficient to ensure adequate power to address the hypothesis being tested?

-Were correct statistical analysis used to support conclusions?

-Are there concerns about ethical or regulatory requirements being met?

Reviewer #1: (No Response)

Reviewer #2: -Are the objectives of the study clearly articulated with a clear testable hypothesis stated? Objectives are not clear and need to be reformulated. 

-Is the study design appropriate to address the stated objectives? Not exactly

-Is the population clearly described and appropriate for the hypothesis being tested? yes

-Is the sample size sufficient to ensure adequate power to address the hypothesis being tested? yes

-Were correct statistical analysis used to support conclusions? yes

-Are there concerns about ethical or regulatory requirements being met? does not apply as the samples are sandfly DNA

Reviewer #3: (No Response)

**Results**

-Does the analysis presented match the analysis plan?

-Are the results clearly and completely presented?

-Are the figures (Tables, Images) of sufficient quality for clarity?

Reviewer #1: (No Response)

Reviewer #2: -Does the analysis presented match the analysis plan? Yes

-Are the results clearly and completely presented? yes

-Are the figures (Tables, Images) of sufficient quality for clarity? yes

Reviewer #3: (No Response)

**Conclusions**

-Are the conclusions supported by the data presented?

-Are the limitations of analysis clearly described?

-Do the authors discuss how these data can be helpful to advance our understanding of the topic under study?

-Is public health relevance addressed?

Reviewer #1: (No Response)

Reviewer #2: -Are the conclusions supported by the data presented? No

-Are the limitations of analysis clearly described? No

-Do the authors discuss how these data can be helpful to advance our understanding of the topic under study? No

-Is public health relevance addressed? Yes

Reviewer #3: (No Response)

**Editorial and Data Presentation Modifications?**

Reviewer #1: (No Response)

Reviewer #2: (No Response)

Reviewer #3: Some minor grammar and spelling suggested edits in attachment.

**Summary and General Comments**

Reviewer #1: In their paper entitled "Performance evaluation of nine reference centres for effective surveillance of Leishmania-infected Phlebotomine sand flies and basis for technical recommendations", Prudhomme & Delabarre et al. present the findings of a multicentre study involving nine reference laboratories. The aim of the study was twofold: firstly, to compare various DNA extraction protocols and, secondly, to conduct a qPCR-based external quality assessment.

The manuscript is of great interest and is well written, although it would benefit from a few minor revisions as suggested. 

Lane 156: it would be useful to include the names of the participating centres in the table.

Lane 157: the PCR protocol, which has been previously described by Mary et al. (2004), is a highly sensitive method. However, no information regarding its specificity is provided in the initial article or in this study. It would be valuable to understand how the authors excluded aspecific amplification and whether they sequenced random samples.

With regard to the comparison of extraction protocols (Lane 219), it would be helpful to know whether the authors assessed the purity and concentration of the DNA obtained through 260/280 nm absorbance measures using a NanoDrop spectrophotometer. This value should be taken into consideration when comparing the performance of the different extraction protocols. I suggest providing further detail on this and to discuss the concerning results in greater depth.

Reviewer #2: Leishmaniasis is a neglected diseases that requires more in-depth research in all directions including diagnosis. This is what make the study important and interesting. However, several comments below should be considered:

Abstract

Line 35: Would it be good to mention the other common name for qPCR, real time PCR?

Line 36: replace the word ‘condition’ by a word related to method of diagnosis or detection.

Line 38-39: In the aim, would better to insert 'DNA in sandflies' somewhere. The title of the manuscript (evaluation of nine centers) and the aim (compare various DNA extraction protocols) should run parallel. It would be helpful to modify one of them to match the other. 

The abstract does not mention sandflies!

The conclusion and significance are general wording that do not convey a real scientific message. 

Author summary

Line 61: Correct 'nfected' to 'infected'

Line 71: add " and proper treatment'.

Introduction:

The introduction should enriched with more studies that compared DNA extraction methods and amplification methods.

Line 107: to conduct EQA, but for what?

Again, the aim in the introduction is different from that in the abstract, one talks about comparing extraction methods while the other about Leishmania detection methods. The title mention evaluation of nince centers. Re-phrase your aim.

Methods:

Line 118: Is the he Rennes Lab the same as INSERM?

Line 120: Why 'the s--called"? Is a reference laboratory or not?

Line 151: PBS is phosphate buffered saline?

Line 154: Do you mean (200 μL of EQA sample)?

Table 1: Handmade means manual work/extraction. May be the later expression is better.

 Tissue lyser II is not an extraction device but rather a disruptor?

Line 158: It will be an added value to justify why you selected this method in particular knowing that we have many good one out there? Is it the method regularly used by all the participating labs?

Line 164: Cq is quantification cycle. While, threshold cycle is Ct. Both are the same.

Line 187: How was the shipment condition monitored? Was there a temperature data logger/indicator? Shipment condition might not be at room temperature, but higher in summer.

Results 

Line 214: Why is L. major Cq is higher than L. infantum? Could be explained in the discussion section.

Line 223: Any explanation why DNA concentration was not determined after the extraction? DNA yield and quality are direct parameters to assess efficiency of DNA extraction method.

Tables 2 and 3: can be put as supplements and turn this table into a table or figure that highlights sensitivity. Cq highly variable between centers. Explain.

Discussion

Line 245: The sentence "The CLIMOS...." is unnecessary.

Line 295: Diagnostic validity includes sensitivity (detecting minimum leishmanial DNA concentration) and specificity (detecting Leishmanial DNA only). Precision (triplicate runs giving close Cq results). This could be discussed to add more depth to the discussion.

Discuss why Cq for L. major is higher than L. infantum.

Line 297: Any explanation why EZI kit performed better than the others?

Line 301: CLIMOS has been acknowledged for funding, which is enough.

Fig 2: High Cq variability between centers ranging from 23 (center 8) to 43 (center 8). Explain. 

Fig 1 and 2 are reflecting the efficiency of the centers in the detecting the leishmanial DNA as a function of Cq value depending on the setup they are using such as the extraction kit, extraction machine, and thermal cycler. This has not been discussed although the title of the manuscript is 'evaluating the nine center...."

Reviewer #3: The paper entails sending a panel of equal Leishmania parasite containing samples to different Leishmania reference labs in Europe to compare extraction techniques effect on ability to detect Leishmania DNA while controlling for qPCR reagents and amplification conditions. The results demonstrate that all centers are able to detect pathogen in the surrogate surveillance samples but also showed the different nucleic acid isolation kits and techniques do introduce certain levels of variation in the Cq of the results. This is important work to do periodically to ensure detection mechanisms are still comparable and sufficient, offer data to support standardization of DNA isolation kits used for epidemiological surveillance purposes and identify centers that need to improve techniques. The study is well designed and straightforwardly presented. Nice work and useful, practical information for the field.

PLOS authors have the option to publish the peer review history of their article (what does this mean?). If published, this will include your full peer review and any attached files.

Reviewer #1: No

Reviewer #2: Yes: Amer Al-Jawabreh

Reviewer #3: No
---

## [Decision Letter · Decision Letter 1]

20 Nov 2024

Dear Dr. PRUDHOMME,

We are pleased to inform you that your manuscript 'Performance evaluation of nine reference centers and comparison of DNA extraction protocols for effective surveillance of *Leishmania*-infected Phlebotomine sand flies: basis for technical recommendations' has been provisionally accepted for publication in PLOS Neglected Tropical Diseases.

Best regards,

Clarence Mang'era, PhD

Guest Editor

Paul Mireji

Section Editor

Shaden Kamhawi

co-Editor-in-Chief

Paul Brindley

co-Editor-in-Chief

Reviewer's Responses to Questions

**Key Review Criteria Required for Acceptance?**

**Methods**

-Are the objectives of the study clearly articulated with a clear testable hypothesis stated?

-Is the study design appropriate to address the stated objectives?

-Is the population clearly described and appropriate for the hypothesis being tested?

-Is the sample size sufficient to ensure adequate power to address the hypothesis being tested?

-Were correct statistical analysis used to support conclusions?

-Are there concerns about ethical or regulatory requirements being met?

Reviewer #1: (No Response)

Reviewer #2: The authors have answered all the points I raised the R1

Reviewer #3: (No Response)

**Results**

-Does the analysis presented match the analysis plan?

-Are the results clearly and completely presented?

-Are the figures (Tables, Images) of sufficient quality for clarity?

Reviewer #1: (No Response)

Reviewer #2: Results are well-presented

Reviewer #3: (No Response)

**Conclusions**

-Are the conclusions supported by the data presented?

-Are the limitations of analysis clearly described?

-Do the authors discuss how these data can be helpful to advance our understanding of the topic under study?

-Is public health relevance addressed?

Reviewer #1: (No Response)

Reviewer #2: Conclusions have been modified as requested in R1

Reviewer #3: (No Response)

**Editorial and Data Presentation Modifications?**

Reviewer #1: (No Response)

Reviewer #2: Accept

Reviewer #3: (No Response)

**Summary and General Comments**

Reviewer #1: (No Response)

Reviewer #2: The Authors have taken all concerns raised in R1 into consideration and made the necessary modification

Reviewer #3: My suggested revisions have been addressed and this paper is ready for publication. Thank you!

PLOS authors have the option to publish the peer review history of their article (what does this mean?). If published, this will include your full peer review and any attached files.

Reviewer #1: No

Reviewer #2: **Yes: **Amer Al-Jawabreh

Reviewer #3: No

---

## [Editor Report · Acceptance letter]

13 Dec 2024

Dear Dr. PRUDHOMME,

We are delighted to inform you that your manuscript, "Performance evaluation of nine reference centers and comparison of DNA extraction protocols for effective surveillance of *Leishmania*-infected Phlebotomine sand flies: basis for technical recommendations," has been formally accepted for publication in PLOS Neglected Tropical Diseases.

Best regards,

Shaden Kamhawi

co-Editor-in-Chief

Paul Brindley

co-Editor-in-Chief
